# Gender Medicine: A New Possible Frontiers of Venous Thromboembolism

Tiziana Ciarambino [1,*,†], Pietro Crispino [2,†], Ombretta Para [3] and Mauro Giordano [4]

1 Internal Medicine Department, Hospital of Marcianise, ASL Caserta, 81025 Marcianise, Italy
2 Emergency Department, Hospital of Latina, ASL Latina, 04100 Latina, Italy
3 Emergency Department, Hospital of Careggi, University of Florence, 50121 Firenze, Italy
4 Advanced Medical and Surgical Sciences Department, University of Campania, L. Vanvitelli, 80016 Naples, Italy
* Correspondence: tiziana.ciarambino@gmail.com
† These authors contributed equally to this work.

**Abstract:** From the data in our possession derived from the literature, it has been shown that men have a high risk and appear to have a specific higher risk of suffering from thromboembolism than women, regardless of age group. Furthermore, at present, this difference between the two sexes has not yet been justified by scientific evidence. Taking this evidence into account, this review was designed to find information in the literature that could be potentially useful and that is crucial to knowledge about the differences between men and women in the genesis of thromboembolic disease. The role of the constitutional and physical factors underlying the difference between the two sexes, the different habitual lifestyles, the role of sex hormones, and what happens in situations such as venous thromboembolism in the course of a neoplastic disease or in the case of pregnancy. We have also focused our efforts on correlating the gender differences between men and women in thromboembolic disease with the dysregulation of the metabolism and the activation of the inflammatory response that often underlies this pathology.

**Keywords:** gender/sex differences; systemic venous thromboembolism

## 1. Background

Pulmonary embolism (PE) and deep vein thrombosis (DVT) constitute the two main manifestations of venous thromboembolism and were considered advanced-age diseases with an annual incidence, according to American data, of at least 120 cases per 100,000 person-years among Caucasians. The estimated incidence in the population at risk, adjusted for age and sex, is higher in men (130 events on 100,000 subjects) than in women (110 on 100,000 subjects). Overall incidence rates in women are slightly higher than in men of the same age, as the two groups are exposed to different extrinsic risk factors. In fact, in younger populations, women have a higher incidence of thromboembolic events linked to hormonal fluctuations that change the ratio between estrogen and progesterone, as occurs during pregnancy and the puerperium [1]. On the other hand, if we consider the two gender groups after an age threshold of over 45 years, however, the incidence of thromboembolic events seems to be higher in men. The causes of this phenomenon are currently not fully understood by the scientific literature. Taken together, the observations of recent years show that the possibility of incurring any thromboembolic phenomenon is a constant and not negligible occurrence throughout life and that this risk remains constant in both men and women, although the literature data agree that men have a slightly higher risk of VTE than that observed in women. Without the identification of those mechanisms, possibly protective in women or vice versa responsible or facilitating in men, it will be difficult to implement health activities aimed at preventing such manifestations, specific to gender. Thrombosis derives from the interaction of three factors: endothelial damage,

venous stasis, and hypercoagulability. There is no thrombotic condition without these risk factors that intervene at different levels of the triad and have different predictive values depending on which they are considered. Risk factors for VTE can be stratified into weak, intermediate, and strong. Weak risk factors are advanced age, obesity, bed immobilization for more than three days, recent laparoscopic surgery, and pregnancy. Intermediate risk factors are include the presence of an active neoplasm, the use of chemotherapy, heart failure, thrombophilia, the use of estrogen-progestogens, and the puerperium. Strong risk factors are major general surgery (orthopedic, oncological, neurosurgical, and major trauma) and atrial fibrillation [1,2] (Figure 1).

| Venous Thromboembolism | | | | |
|---|---|---|---|---|
| *Endothelial Alterations* | | *Hypecoagulable State* | | *Venous stasis* |
| **Damage** | **Dysfunction** | *Acquired* | *Hereditary* | *Acquired* |
| Venous cathehter | Smoking | Tumors | Mutation factor V Leiden | Immobility |
| Surgery | Hypertensions | Chemotherapy | Mutation prothrombin gene | Hemoconcetration |
| Trauma | Autoimmune infiltrate | Radiotherapy | Proteina C and S Deficency | Policytemia |
| Infusions | Chronic persistent inflammation | Pregnancy | Antithrombin III deficency | Emolisys |
| | Cirrhosis | Obesity | | |
| | | Thrombocitosis | | |

**Figure 1.** Classification of venous thromboembolism.

## 2. Men and Women Regarding Thromboembolic Risk

Up to the age of 70, a gender difference was observed in favor of men rather than women, which relates to the intrinsic risk of VTE (without considering possible extrinsic or precipitating factors). On the other hand, the higher incidence of VTE observed in women during the fertile age is largely due to transitory or common phenomena limited in time such as pregnancy and the puerperium, the use of estrogen-progestin drugs used for contraceptive purposes, or as therapies for the prevention of other genital diseases that are common in childbearing age [3]. The possibility of incurring a thromboembolic event is significantly higher in pregnant women than in non-pregnant women due to the phenomena that regulate the implantation of the trophoblast in the uterus and, subsequently, to those that regulate the detachment of the placenta. This risk of PE is five times higher in the puerperium when there are major coagulation phenomena in the third trimester of pregnancy, in particular, the increase in coagulation factors and the reduction of natural anticoagulants that persist beyond delivery [4]. As for estrogen-progestins, pregnancy and the puerperium also have a synergistic action with acquired and hereditary thrombophilia defects, such as antithrombin III deficiency and proteins C and S, although the presence of these alterations is rare [5]; congenital defects of antithrombin, protein C, and protein S are together responsible for about one-fifth of thrombotic episodes in young subjects under the age of 50. Furthermore, it should be noted that due to the genetic phenomenon of the incomplete penetrance of these defects, not all deficient individuals develop thrombosis. In the subject with the absence of proteins C and S deficiency, higher concentrations of protein C and protein S were observed in men than in women, but qualitatively, in women, the mean protein C and protein S activity increases with age. In men, it has been observed that the average levels of protein C increase up to the age of 50 and subsequently tend to decrease, while the levels of protein S decrease after the age of 50. Antithrombin III levels, on the other hand, remain stable over time in women; instead, they tend to decrease significantly after 50 years of age in men [5]. The absolute risk of thrombosis is high and rises significantly in the post-partum period compared to the previous period [6]. Most of the studies [7–11] agree in considering young and fertile women more at risk of thromboem-

bolic events and consider, on the contrary, that men over 50 years of age are more exposed to thromboembolic risk compared to women, although after age 70, the incidence increases equally in both genders, and, for some studies, the incidence in women may be higher than in men [7,9,11]. However, the data available for older people are insufficient, and only recently, in a sham study that also considered the risk of death associated with thromboembolic events, the incidence of the first episode of VTE was slightly higher in women than men [7]. Therefore, a kind of "catch-up" effect in women seems to be appreciated from these data, which is probably explained by concurrent risks (e.g., death from other causes in men or shorter life expectancy in men than in women) [7]. Pulmonary embolism is the first manifestation of VTE and most frequent in women, while deep vein thrombosis prevails in men. Since the difference is more significant when considering individuals with idiopathic VTE, it follows that the fact that the woman does not have extrinsic risk factors specific to the female gender but is a phenomenon linked to the female gender itself. Unfortunately, even this phenomenon lacks a correct and convincing explanation. It must therefore be remembered that, as regards pulmonary embolism, the diagnosis must be taken into consideration even without clinical signs of peripheral vein thrombosis, with greater attention given to women. Furthermore, data available in the literature show that men with pulmonary embolism more frequently present with symptoms of onset episodes of haemoptysis and chest pain, while women more often have dyspnoea or syncope [12,13].

## 3. Factors Involved in Venous Thromboembolism

VTE is a disease that has a complex and multifactorial genesis, as it occurs because of the interaction between acquired factors and hereditary factors that make it possible to establish a pro-thrombotic state in the body [14].

### 3.1. Acquired Predispositions to Thrombosis

As we have seen, environmental and acquired factors play a determining role in the incidence of VTE in the two different genders [15]. The presence of these various conditions of disability, the presence of neoplasms and chronic diseases, and various lifestyles could contribute to the different incidence of VTE that exists between the two sexes [14–19]. Regarding immobilization and all the conditions related to it, Heit et al. [15] estimated that VTE is very common in men, manifesting mainly as deep vein thrombosis of the lower limbs, while, on the contrary, a subsequent study showed no difference in the two genders [20]. Contrary to what was previously said, a study [21] investigated the link between cigarette smoking and VTE in young men and women, attempting to demonstrate a positive association between smoking and thromboembolic events in men and women. Although a synergistic association was found between smoking and VTE in middle-aged men and women, it was also noted that in women, this association was valid for all doses of tobacco, while in men, the association was only significant for the consumption of high doses of tobacco. Heavy smokers were shown to have a higher risk of pulmonary thromboembolism than men with moderate tobacco use, although reducing tobacco use was not effective in reducing the risk of VTE in this category of subjects. It is well known that there are evident physical differences between men and women, especially as regards the anthropometric component (height and weight) and body composition, including the distribution of fat in the various parts of the body. Men are, on average, taller than women. Greater body height has been identified as a risk factor for VTE in both men and women [22,23]. Furthermore, a sedentary lifestyle and all those conditions that induce venous stasis are considered elements that increase the risk in people of tall stature [24,25]. However, considering only height as an important factor in the determinism of thromboembolic phenomena, a complete explanation of any gender differences is not obtained. Further studies are needed to study how the different anthropometric parameters, such as height, obesity, quantity, and different distribution of visceral fat within the liver parenchyma, combine with each other and can elicit variations in the body's metabolism and consequently promote an inflammatory state that is always accompanied by haemocoagulative phenomena and

finally with an increase in thromboembolic risk. Abdominal obesity, which has always been considered a risk factor for cardiovascular disease for both men and women, causes an increased risk of thrombosis, twice that of thin subjects, because abdominal fat is associated with a greater formation of thrombin and therefore a greater pro-thrombotic activity [26]. Steffen et al. [26] observed that the metabolic syndrome occurs concomitantly with a higher risk of idiopathic VTE, especially in men rather than women, precisely because abdominal obesity is more easily absorbed in the former than in female subjects. High body mass index and obesity are, for this reason, now considered consolidated risk factors for VTE [27]. The increase in total body fat and the amount of visceral fat frequently leads to increased levels of procoagulant factors [28]. Inflammation and metabolic dysregulation, and the high concentration of coagulating factors intertwining with each other, play a key role in the pathogenesis of VTE and have recently received much attention [29]. Although classically VTE and arterial thrombotic events have been considered separate diseases, it has been established that the two entities share several risk factors, such as chronic inflammatory status [30]. Although the mechanisms by which inflammation is associated with systemic thromboembolism have not yet been elucidated, the chronic inflammatory state underlies any clinical reasoning in all those patients with or at high risk of VTE. If the chronic inflammatory state has been associated with an increased risk of VTE, all acute and intense inflammatory states are also closely linked to an increased risk of VTE. Furthermore, it has been found that, together with elevated hepatic triglyceride concentrations, there is an increase in coagulation factor IX [31]. In another study, the prevalence of VTE in patients with nonalcoholic fatty liver disease was approximately three times higher than in controls [32], and women appeared to be less exposed to thromboembolic events because, unlike men, they have a higher total body fat concentration, but the proportion of visceral and hepatic fat is lower than in men [33,34]. Unfortunately, we must underline that while both the high and low-grade inflammation observed in obese subjects, or more generally in those affected by the metabolic syndrome, is relevant in the pathogenesis of VTE, such as arterial thrombotic disease, its role is largely unknown and not yet studied [34].

### 3.2. Acquired Predispositions to Thrombosis and Reproductive Factors in Women

After this, it was said that men have more risk factors for developing VTE than women. As for women, the use of oral contraceptives, the need for hormone therapy and therapy with the selective estrogen receptor modulator, pregnancy, and the postpartum period should be cited as risk factors for VTE. For example, the use of injectable depot-medroxyprogesterone acetate for contraception has been associated with a three-fold greater risk of venous thromboembolism, while alternatively, the placement of a levonorgestrel-based intrauterine device carries no risk. Hormone therapy considerably increases the risk of VTE, although it varies according to the type of estrogen used and the route of administration, ranging from a high risk during systemic administration to a negligible or no risk with transdermal therapy [35]. The overall incidence of VTE associated with pregnancy is about 200 in a population of 100,000 women. Compared with non-pregnant women of childbearing age, the risk of VTE increased at least fourfold. The risk of VTE during the postpartum period is five times greater than the risk of thromboembolic events during pregnancy. The presence of a previous event of superficial vein thrombosis is an independent risk factor for VTE both during pregnancy and during the postpartum period [36].

### 3.3. Inherited Predispositions to Thrombosis

The principal inherited predisposition factors to VTE may be due in large part to factors depending on genetic characteristics and to factors depending on the concentration of sex hormone levels. Venous thromboembolism is known to be associated with a certain genetic predisposition. The most common genetic alterations in the genesis of thromboembolism are mutations involving the factors that regulate the blood coagulation cascade. The mutation of the factor V Leiden gene, expressed in both homozygous and heterozygous

forms, the mutation in the prothrombin gene, the genetically determined deficiency of protein C and protein S, and the mutation affecting the gene encoding antithrombin III are determinants of VTE that have been recognized [37,38]. It has been observed that all blood groups other than group O also have a certain predisposition to VTE. However, despite all this evidence, it is clear, first, that it is not possible to derive gender differences in the risk and in the greater severity of VTE from these mutations [39]. Unlike these mutations, it has been hypothesized that mutations in the genes coding for coagulation factors VIII and IX, associated with an increased risk of VTE when present at high levels, could lead to sex-specific changes as the site of the mutation is on the X chromosome. In practice, however, the studies obtained from a large series of patients followed for thromboembolic risk have not provided useful explanations for the different predispositions to VTE in the two sexes [40–43]. The mutation of the normal haplogroup of the male sex chromosome Y has also been called into question, but even in this case, it is considered unlikely that this mutation could be the basis of the difference in the risk of new VTE or recurrent VTE between women and men [44]. Sex hormones have also been attributed to an increase in the risk of VTE, although the data available in the literature are few and discordant. It would therefore seem that between the two genders, there is no association between estrogen and testosterone levels and the risk of VTE [45]. There was only a relationship between sex hormones in pregnant women and the risk of VTE during the nine months of gestation. In fact, with the progress of pregnancy, in parallel with the levels of estrogen and progestin, the concentrations of clotting factors tend to increase, and with them the risk of VTE [46]. Another situation quite different from the physiological one of pregnancy, in which sex hormones play a prothrombotic role, is the pathological one of polycystic ovary syndrome (PCOS). PCOS is a syndromic disease that affects mostly young patients who show all the characteristics of hyperandrogenism, a marked ovulatory dysfunction, and alterations in the morphology of the ovary ova the normal parenchyma is ecstatically replaced by multiple cystic formations [47,48]. It has been observed in several studies that the androgen excess common to this sex-specific pathology is associated with higher concentrations of pro-coagulate and pro-inflammatory factors [49,50]. Considering the evidence regarding female subjects outside of pregnancy and PCOS, little is known about the relevance of the role played by sex hormone levels in determining the risk of VTE. From the available data, it can be concluded that there is a moderately increased risk of VTE in all female subjects with higher estrogen levels and in all cases of hyperandrogenism found in young women. Hyperandrogenism appears to be an important endocrinological disorder, also considering the well-demonstrated risk of major thromboembolic events in men, although higher testosterone levels are not the only contributors to the greater risk of VTE in men. In fact, we have underlined how the common hyperandrogenism in young women during PCOS or pregnancy is also associated with the activation of factors of inflammation [51]. Whether the synergism between pro-coagulation activity and inflammation is a common mechanism also in males and is responsible for the increased VTE risk observed in this group of patients has yet to be demonstrated in the male subject.

Sex Differences in Endothelial Function

The endothelium plays a fundamental role in the biological processes that guarantee the physiological integrity of the vessel, synthesizing numerous substances capable of modulating the tone of the vessels, the inflammatory and immune systems, and platelet function [52]. A functional alteration of the endothelium, which occurs, for example, in the presence of the main cardiovascular risk factors, reduces this protective role so that the endothelium synthesizes and releases fewer protective substances, and, moreover, it can turn into a harmful organ as it is induced to synthesize substances with a vasoconstrictive, pro-aggregating, and pro-inflammatory action [52]. Physiologically, the endothelium is defined as a barrier in which an appropriate balance is present between vasoconstrictor factors and endothelial vasodilators. This balance is important for vascular homeostasis. On the contrary, the reductions in endothelial function are detrimental and predict and

precede the development of overt cardiovascular events [52]. A healthy endothelium is characterized by an appropriate release of nitric oxide (NO) by vascular endothelial cells in response to a vasodilator stimulus. Gender differences have been reported regarding the regulation of endothelial function in releasing NO [53]. These differences consist mainly in the functioning of intracellular signal transduction systems of angiotensin and endothelin-1 (ET-1) [53,54]. These mechanisms, which contribute to sex differences in endothelial function in healthy young women and men, may explain why women are protected in premenopause while men have a higher risk of cardiovascular events already at a young age [55]. Sex hormones affect endothelial function. In women, estrogen receptors (ERs), once activated, promote NO release by endothelial activation of NO synthase (eNOS), while androgen receptor engagement can cause endothelial NO release. In men, androgens can induce increases in blood pressure through mechanisms that include reactive oxygen species (ROS) [55]. However, androgens also play a cardioprotective role in men through direct conversion to or effects on estrogen [55]. There are also sex-specific differences related to sex hormone-mediated endothelial signal transduction, such as those regarding the function of the renin-angiotensin system and ET-1 [55]. These sex differences are, directly and indirectly, mediated by differences in the gonadal hormonal environment, which lead to sex differences in the expression and/or activity of proteins at the level of endothelial cells and vascular smooth muscle cells [56]. However, aging and the concomitant reduction of sex hormones favor the loss of cardiovascular protection in women compared to men of the same age after menopause [57]. Overall, gender differences in mechanical control and in maintaining endothelial function play an important role in the pathogenesis of cardiovascular diseases, and therefore identifying gender-specific elements for the treatment or prevention of these conditions is essential for a greater containment of morbidity and mortality of both sexes at every stage of life [52].

## 4. VTE Outcomes and Gender Differences

Some retrospective studies [58,59] that have analyzed the outcomes of patients have shown that even hemodynamically stable female patients have greater mortality both in the hospital and one month after admission, probably due to more advanced age and comorbidities but also due to differences in thrombotic and fibrinolytic activity and by extension of the disease in the lung. Furthermore, the reduced hospitalization may also be due to the tendency to underestimate signs and symptoms attributed to anxious states, especially at a young age with a delay in diagnostic-therapeutic timing [60]. As for anticoagulant therapy, the data are conflicting. Second, in some studies, women seem to be subject to a greater risk of bleeding, even with the new oral anticoagulants, due to the different pharmacokinetics of the two sexes, mainly linked to a different distribution of adipose tissue, different creatinine clearance, and a different volume of distribution. different drug absorption, different plasma protein binding, and differences in urinary excretion [61,62]. In a 2015 meta-analysis, Dentali et al. [63] demonstrated the safety of the use of anticoagulants in patients with VTE and non-valvular atrial fibrillation without finding gender differences in the incidence of adverse events. On the other hand, in a meta-analysis [64], the difficulty in analyzing data relating to the risk of bleeding in women was highlighted, as there are significantly lower percentages of patients on new oral anticoagulants in the latter. However, another study pointed out that there is a high incidence of side effects in women even with the use of low-molecular-weight subcutaneous heparin, especially when administered in high doses [65]. In the case of PE, hemodynamic status at presentation has been considered the most important prognostic factor to determine short-term mortality [66]. As for the risk of a subsequent thromboembolic event after the first episode of VTE, although we have so far listed the differences between the two genders, it was observed that both groups had a similar rate of relapse over a one-year observation period. Furthermore, once the first episode of acute VTE was overcome, it was observed that women, when interviewed with appropriate clinical questionnaires, had scores of quality of life and satisfaction with the treatment obtained that were lower than expected than those of their male counterparts [3].

In about 30% of patients who suffered from a first thromboembolic event, a recurrence was observed within 10 years of the first event [15]. From the data available in the literature, it can be observed that the risk of relapse is not always the same but decreases from year to year, varying from a higher risk common in the first 6–12 months to a much lower risk when considering the following years. The risk of relapse, however, may never tend to cancel itself out. Furthermore, patients who suffer from recurrent thromboembolism are significantly more possibilities to have the same presentation pattern as the first VTE event. The main predictors of future recurrence of VTE include an increase in the patient's age, a higher body mass index, male sex, the concomitant presence of an active neoplastic pathology, and conditions of permanent immobility linked to disabling diseases of the lower limbs. or neurological on a traumatic or degenerative basis. Additional predictors of relapse include all forms of idiopathic VTE or those related to the presence of elevated circulating levels of anticoagulant lupus or antiphospholipid antibodies, antithrombin III deficiency, or genetic deficiency of protein C or protein S, or increased plasma homocysteine levels, or a persistent increase in plasma D-dimer, which is a frequently observed abnormality and usually found in patients with idiopathic VTE or where it is possible the presence of a residual venous thrombosis. In cancer patients with active or uncontrolled cancer, the risk of recurrence of VTE may be high depending on the location of the primary tumor. Neoplasms such as pancreatic cancer, neoformations of the brain, lungs, and ovaries, various myeloproliferative or myelodysplastic disorders, and all advanced stage IV cancers associated with concomitant situations of disability are at increased risk of thromboembolic complications [67]. The role played by advancing age in VTE recurrence is not yet fully understood, but thromboembolic events are unrelated to the patient's age, since older age and younger age are the groups most associated with the risk of recurrence mainly among women [68], while in another study [69] there was no gender difference considering the patient's age during relapse. In cancer patients with active disease and concomitant VTE, male sex, location, an advanced form of cancer, and a previous history of VTE were considered predictors of VTE recurrence despite anticoagulation therapy [70]. Among active cancer patients with VTE, patient sex, cancer site (lung, breast), cancer stage, and previous VTE were predictors of VTE recurrence during anticoagulation therapy [71]. All these data, taken together, suggest that VTE may, in most cases, be a disease limited to a single episode, but in some cases, it may potentially be a chronic disease with episodic recurrence.

## 5. Superficial VTE and Gender

Although superficial vein thrombosis (SVT) of the lower extremities has traditionally been considered a relatively benign disease, there are studies in the literature suggesting a concomitant pattern of DVT or PE among these patients, highlighting that the risk of subsequent DVT or PE during follow-up is not negligible [72–76]. However, the prevalence of deep venous system involvement in patients with SVT is highly variable across different studies. In a meta-analysis [77], summarizing data from 22 studies, it was noted that risk factors, including older age, obesity, cancer, prior thromboembolic episodes, pregnancy, oral contraceptives, hormone replacement therapy, recent surgery, and autoimmune disease, are common to both DVT/PE and SVT. The prevalence of DVT/PE in SVT patients has also been observed to be lower in younger subjects, female patients, recent trauma patients, and pregnant women. These results are in agreement with data from previous large studies [74,78]. This meta-analysis [77] finds no relationship regarding SVT associated with the presence of varicose veins in the lower limbs with major thromboembolic events and therefore does not highlight gender differences in this phenomenon. Currently, there are no prognostic prediction models for the risk stratification and management of SVT in clinical use, making its clinical management ambiguous and uncertain, potentially putting patients at risk of major thromboembolic complications who are erroneously left without anticoagulant therapy or, instead, putting patients unnecessarily at risk of bleeding complications by adopting antithrombotic treatment [76,79]. It would seem that in the risk assessment models of major thromboembolic events, both previously and at present,

the role of gender is not contemplated as a prognostic marker capable of optimizing the pharmacological treatment of the superficial thromboembolic disorder.

## 6. VTE, Cancers, and Gender

Venous thromboembolism (VTE) contributes to up to 20% of cases to complicate the general health status in cancer patients, representing a commonly diagnosed condition that significantly increases the morbidity and mortality rates of this category of patients. Cancer patients are, in fact, in a persistent prothrombotic state, linked to alterations of the hemostatic-coagulant system induced by the presence of the neoplasm, stasis, and slowing of blood flow induced by perilesional edema, especially in advanced forms, endothelial dysfunction, vascular inflammation, and activation linked to the metastatic power of the neoplasm which uses angiogenetic, pro-inflammatory, and platelet-activating factors to reach the venous vessels. The increased risk of VTE in the neoplastic patient is generally related to the patient's characteristics, the tumor, the degree of progress, and/or the treatment performed, which, at various levels and titers, can contribute to the Virchow triad, thus altering the mechanisms of hemostatic agents that balance thrombosis and lysis of the clot and, therefore, to determining a state of hypercoagulability [80]. Nasser et al. [81] proposed an interesting stratification of the cancer-related hypercoagulability state into two main entities; type 1 hypercoagulability, deriving from the degradation of endogenous heparin by heparinase secreted by tumor cells with metastatic capacity, and type 2 hypercoagulability, which instead includes all those other previously discussed and common causes during neoplastic pathology. Heparinase is not only a key enzyme capable of altering the internal balance present by pro and anticoagulant factors, but in fact, it has also been shown to play its role in degrading the low molecular weight heparin and heparin that are commonly used in the prophylaxis and treatment of venous thromboembolism in these patients, with consequent neutralization of the pharmaceutical properties of these molecules. It is interesting to note that heparinase is highly expressed, especially in some cancers such as those with lung, gastric, and pancreatic localization. This would help explain why some cancers are at increased risk for VTE. However, gender differences in cancer-related VTE are not yet fully defined because many of the studies that have addressed this problem have been conducted in patient groups with unclear or non-homogeneous inclusion criteria [82]. Ultimately, still today, the correct VTE prophylaxis is decided for each individual patient by adjusting the dosage of therapy according to the risk and the number of risk factors. This strategy has been shown to be effective in reducing the recurrence of VTE events [82]. The evidence-based clinical practice guidelines produced so far do not make a gender distinction in the diversification of recommendations for the treatment of VTE in cancer patients or for the prophylaxis of VTE in cancer patients [83]. However, in some cases, it is necessary to think from the perspective of gender medicine, because in making the most appropriate therapeutic choice, it is necessary to consider that some neoplastic conditions are exclusive to the female sex or because it is necessary to consider the various periods of life that mark the life of the subject's female [83]. For example, it is known that postmenopausal women with breast cancer cannot fully benefit from tamoxifen hormone therapy as this agent is associated with an increased risk of VTE, while the aromatase inhibitors offered as an alternative can be used successfully as they are characterized by a high efficacy in contrasting tumor recurrence and a low risk of VTE. It has also been observed that women with gynecological malignancies are at high risk of developing VTE, yet prophylactic anticoagulant treatment is currently little used in these patients [83,84]. The presence of cancer alone may be sufficient to indicate thromboprophylaxis in pregnant patients at high risk of VTE, although, prophylactic treatment is often adapted to the characteristics of the tumor, the stage of progression of the neoplasm, and the gestational period, as well as to informed consent to prophylactic therapy [85,86]. Currently, the various formulations of low molecular weight heparin for subcutaneous use are recommended for the treatment of VTE in these women because they do not cross the placental barrier, unlike vitamin K antagonists and new direct-acting oral anticoagulants, which are currently contraindicated

because they can reach the fetus [84,86,87]. It is interesting to note that some patients who present to clinical observation with signs and symptoms of full-blown VTE have a certain risk of having an occult neoplasm for which thromboembolism represents its first indirect manifestation even before the neoplasm directly shows the signs of its presence and invasiveness. In this regard, it is interesting to consider the results of a study [88] that demonstrated the presence of this phenomenon mainly in males. Although minimally, this finding could also be partly responsible for the higher risk of VTE in men than in women. Furthermore, this data depends on the type of cancer diagnosed. Men with VTE had a higher risk of developing an occult cancer type in the course of their own or more frequent malignancies in the male gender, while among women with VTE, the probability of finding an occult neoplasm was lower [88]. In these cases, the type of cancer diagnosed as underlying VTE is not always prevalent in women, except for gynecological cancers. The same study shows that this phenomenon was not related to the degree of invasion and the stage of the hematological disease [88].

*VTE and COVID-19 Infection*

Numerous studies have shown an association between thrombosis and infections [89,90] with very conflicting results. In the case of COVID-19 infection, the indication between the severity of the infection and VTE is greater, and this could be explained by several pathophysiological alterations typical of this virosis, and which foresee a direct effect of the virus on endothelial cells, which, in turn, causes an exaggerated inflammatory response, downregulation of angiotensin 2 converting enzyme receptors, and activation of the coagulation system [91,92]. Although deep vein thrombosis and pulmonary embolism traditionally belong to the disease spectrum of venous thromboembolism, the relative incidence of pulmonary embolism was much higher, which could be due to immunothrombosis (thrombosis in the pulmonary vessels from local inflammation) [93]. The increased bleeding risk may be related to endothelial dysfunction, coagulopathy, or disseminated intravascular coagulation [94,95].

On the other hand, considering the risk of venous thromboembolism after COVID-19, the studies conducted so far have shown conflicting results [90–92]. Although a meta-analysis [96] reported an incidence of venous thromboembolism of approximately 13%, mainly including patients with severe COVID-19 during the first wave of the pandemic. Another report, including studies with a control group design, did not show an increase in the rate of venous thromboembolism [97]. This study found an increased risk of a first deep vein thrombosis up to three months after COVID-19, pulmonary embolism up to six months, and a bleeding event up to two months, with the risk of pulmonary embolism in the acute phase particularly high.

From reports evaluating the gender difference in COVID patients with manifestations of venous thromboembolism, it would appear that sex does not have a significant effect on the association between COVID-19 infection and deep vein thrombosis. As regards pulmonary embolism in the course of COVID-19 virus infection, a significant gender difference was found; in particular, the incidence rate ratios were higher in male than female participants during the first three months after COVID-19 and higher in the 50–70 age group [97]. Cerebral vein thrombosis (CVT) is an extreme manifestation of venous thromboembolism, as it can represent a real, uncommon neurological emergency associated with significantly high morbidity and mortality and is difficult to diagnose as it is often not possible to differentiate it from other conditions [98]. The CVT can present a large variety of clinical pictures, but the most common symptom is headache, followed by focal neurologic deficit, seizure, and altered mental status. Among CVT, cerebral venous sinus thrombosis (CVST) is more common in women than in men, possibly due to gender-specific risk factors in young adults [98]. This condition is due to thrombosis of the cerebral veins, which results in obstruction of the venous outflow and increased intracranial pressure [98]. Women under the age of 50 are most affected and risk factors for the condition include pregnancy, use of drugs (oral contraceptives), hereditary thrombophilic status, previous venous thromboembolic

events, malignant tumors, severe infections, and recent neurosurgery [98]. A study has shown that depending on the number of female-specific risk factors during cerebral venous thrombosis (CVT), there can be differences in imaging parameters and neurological severity. In fact, it was observed that women with an isolated risk factor had a better radiological picture and a better prognosis in the following months. There was no difference between the risk factors common to both sexes except for the state of hyperhomocysteinemia, which was significantly more frequent in males. As regards the clinical presentation, the finding of disturbances of the state of consciousness and the sensory system was significantly higher in females than in males. Mortality rates, on the other hand, did not differ according to gender or according to the number of risk factors. Cerebral venous thrombosis with an isolated risk factor is a rare event and is not characterized by gender differences in terms of clinical-radiological severity and outcome [96–98].

CVT, although rarely, can present to complicate the clinical history of patients with myeloproliferative neoplasms (MPN). However, this relationship between the two pathologies is important because the thrombotic picture could represent the initial manifestation of the underlying hematological disease, but overall, a full-blown MPN. It would appear that both conditions share the JAK2V617F mutation, and as evidenced by a study by the European Leukemia Net (ELN), there is a close association between MPN-associated CVT and the JAK2V617F mutation, younger age, and female gender. This mutation involves increased activity of the JAK2 tyrosine kinase protein, and its role is very important in cell division and even more so in the production of blood cells by the bone marrow. Regarding this mutation, the analysis of the data available in the literature, especially from the last ten years, does not currently reveal gender differences in the prevalence of the JAK2V617F mutation [98].

Recently, an increased incidence of CVT was observed in the post-authorization surveillance phase of COVID-19 vaccines, including unusual cases of thrombocytopenia with thrombosis reported in recipients of adenoviral vector vaccines. One of the devastating manifestations of this syndrome was cerebral venous sinus thrombosis (CVST). The pathogenesis of this clinical form seems to be linked to thrombocytopenia and consequent thrombosis with the presence of autoantibodies against platelet factor 4 (PF4) induced by these vaccines. Based on initial reports, female gender and age less than 60 years were identified as possible risk factors for CVT associated with adenoviral vector vaccines [96–98].

## 7. Conclusions

To conclude, men have a higher intrinsic risk of VTE than women, regardless of age. To date, no studies have been able to fully explain the causes of this difference. Body height and its combination with other anthropometric actors is an established risk factor for VTE but appears to contribute only partially to increasing the risk of thromboembolic events in men. Inflammation and metabolic dysregulation are important determinants of arterial thrombotic risk and appear to be of interest regarding the mechanisms underlying VTE. Overall, little is known about the relevance of sex hormone levels to the risk of (sex-specific) VTE, and from what has been said, it seems that they only play a role in specific conditions common to the female gender. During neoplasia, instead, it would seem that venous thromboembolism is more prevalent in male subjects. A better understanding and a greater extension of knowledge from the perspective of gender medicine could translate into better strategies for the prevention and management of thromboembolic pathology. When women develop a venous thrombotic episode, they have a higher probability than men of presenting with pulmonary embolism in the absence of clinical signs of deep venous thromboembolism, which are more common in men. Regarding pulmonary thromboembolism, the clinical presentation mode is most frequently experienced by men with chest pain and haemoptysis, while women with syncope are often linked to hemodynamic instability. In fact, women have higher in-hospital and 30-day mortality rates due to their more advanced age at onset and comorbidities. Women are more prone to major bleeding during therapy due to different pharmacokinetics between the two sexes, but on the other hand, they must

undergo prophylaxis due to a greater risk of pulmonary embolism. In the future, it will be necessary to identify which factors, genetic or acquired, in both sexes, contribute to increasing the evidence of pulmonary thromboembolism, so as to be able to establish, more accurately, appropriate management and therapeutic pathways that take gender differences into account.

### 8. Take Home Messages

- Men tend to have a greater risk of having thromboembolism than women;
- At present, the gender differences regarding VTE are not well clarified;
- The relevance of sex hormones in the pathogenesis of VTE is recognized only in some typical situations of the female gender;
- In women, syncope is the condition of PTE onset often linked to hemodynamic instability;
- Women with PTE have higher in-hospital and 30-day mortality due to the advanced age of onset and comorbidities;
- Some cases of idiopathic VTE have revealed occult cancers at follow up;
- Women are predisposed to major bleeding during anticoagulant therapy due to different pharmacokinetics in comparison to men;
- It will be necessary to identify which factors, in both sexes, contribute to VTE.

**Author Contributions:** Conceptualization, T.C. and P.C.; methodology, T.C.; software, O.P. validation, T.C., P.C. and M.G.; formal analysis, T.C.; investigation, P.C.; resources, M.G.; data curation, T.C.; writing—original draft preparation, T.C. and P.C.; writing—review and editing, T.C.; visualization, T.C.; supervision, T.C.; project administration, M.G.; funding acquisition, P.C. All authors have read and agreed to the published version of the manuscript.

**Funding:** This research received no external funding.

**Data Availability Statement:** Not applicable.

**Conflicts of Interest:** The authors declare that haven't conflict of interest.

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
