# Peer review of "Gender Medicine: A New Possible Frontiers of Venous Thromboembolism"

_stresses, doi:10.3390/stresses3010013_

Round 1
Reviewer 1 Report
It contains quite broad data and description of the facts in detail. Due to the form of the paper – review and the life-threatening complication in the form of VTE, I have no further comments. From my point of view, the manuscript can be published in the current form.
Author Response
28 November 2022
Dear Prof. Doctor
Please, find enclosed the revised version of the manuscript entitled: “Gender Medicine: A New Possible Frontiers of Venous Thromboembolism” We thank the Editor and the Reviewers for their comments and we hope that the following changes will now make the manuscript suitable for publication on the STRESSES. Please see the following list of the changes made in manuscript.
Reviewer 1.
Query: It contains quite broad data and a description of the facts in detail. Due to the form of the paits – review and the life-threatening complication in the form of VTE, I have no further comments. From my point of view, the manuscript can be published in the current form.
Response: According to the reviewer's suggestion no further changes are needed
Best regards
Tiziana Ciarambino
28 November 2022
Dear Prof. Doctor
Please, find enclosed the revised version of the manuscript entitled: “Gender Medicine: A New Possible Frontiers of Venous Thromboembolism” We thank the Editor and the Reviewers for their comments and we hope that the following changes will now make the manuscript suitable for publication on the STRESSES. Please see the following list of the changes made in manuscript.
Reviewer 1.
Query: It contains quite broad data and a description of the facts in detail. Due to the form of the paits – review and the life-threatening complication in the form of VTE, I have no further comments. From my point of view, the manuscript can be published in the current form.
Response: According to the reviewer's suggestion no further changes are needed
Best regards
Tiziana Ciarambino
Reviewer 2 Report
1 1. I really do not understand why the authors have a separate section on Cerebral venous sinus thrombosis and gender, especially after the Conclusion of the manuscript. If they are going to do this, why not have a separate section on unusual places for venous thrombosis, such as mesenteric, portal, etc.? If they wish to place this in the section on COVID-19 and thrombosis and emphasize the vaccine story, that makes sense to me. However, the way it is now, this does not make sense to me.
2. Would the authors be able to add a section on superficial thrombophlebitis and gender? This would make this contribution even more complete.
3. I found the section on “Factors involved in venous thromboembolism” very difficult to read. I would suggest that the authors divide this section into two or three subheadings…right now, there is only one subheading.
4. The English usage and grammar should be reviewed carefully, as there are a number of areas that could be improved and, there are errors that need to be corrected (ex., page 11, line 535, the word “ad” should be “at”).
5. Page 2, lines 51-55 are very confusing to me. If the authors could even list them, it makes more sense to me….
Weak risk factors:………
Intermediate risk factors:……
Strong risk factors:………..
6. The first paragraph under “Factors involved in venous thromboembolism” seems redundant to what has already been discussed in the “Background”.
7. On page 5, line 187 does not make any sense….”…men have more risk factors for developing VTE than men.”
8. Line 207-208 on page 5, the word “various” is used three times in the same sentence. This should be improved.
9. Line 213 on page 5, the word “zero” should be “blood group O”.
10. On page 7, lines 333 and 334, I did not understand the “o” placed in front of “genetic deficiency…..”, increase in plasma……”, and “persistent increase in……”. Please explain.
11. Line 374, page 8, there is an extra “.” in the line after the word “used”.
Author Response
28 November 2022
Dear Prof. Doctor
Please, find enclosed the revised version of the manuscript entitled: “Gender Medicine: A New Possible Frontiers of Venous Thromboembolism” We thank the Editor and the Reviewers for their comments and we hope that the following changes will now make the manuscript suitable for publication on the STRESSES. Please see the following list of the changes made in manuscript.
Reviewer 2.
- Query: I really do not understand why the authors have a separate section on Cerebral venous sinus thrombosis and gender, especially after the Conclusion of the manuscript. If they are going to do this, why not have a separate section on unusual places for venous thromboses, such as mesenteric, portal, etc.? If they wish to place this in the section on COVID-19 and thrombosis and emphasize the vaccine story, that makes sense to me. However, the way it is now, this does not make sense to me.
Response: According to the reviewer's suggestion, the part dedicated to thrombosis of the cerebral venous sinuses has been included in the section concerning covid 19
- Query: Would the authors be able to add a section on superficial thrombophlebitis and gender? This would make this contribution even more complete.
Response: According to the reviewer's suggestions a section dedicated to gender differences for superficial thrombophlebitis has been added to the text.
- Query: I found the section on “Factors involved in venous thromboembolism” very difficult to read. I would suggest that the authors divide this section into two or three subheadings…right now, there is only one subheading.
Response: According to the reviewer's suggestion the section “Factors involved in venous thromboembolism” has been divided into three subheadings.
- Query: The English usage and grammar should be reviewed carefully, as there are a number of areas that could be improved and, there are errors that need to be corrected (ex., page 11, line 535, the word “ad” should be “at”).
Response: According to the reviewer's suggestion errors in the text have been corrected in the revision.
- Query Page 2, lines 51-55 are very confusing to me. If the authors could even list them, it makes more sense to me….
Weak risk factors:………
Intermediate risk factors:……
Strong risk factors:………..
Response: According to the reviewer's suggestion the texts have been simplified and structured subdividing risk factors into weak, intermediate, and strong
- Query: The first paragraph under “Factors involved in venous thromboembolism” seems redundant to what has already been discussed in the “Background”.
Response: According to the reviewer's suggestion the first paragraph under “Factors involved in venous thromboembolism” has been simplified.
- Query: On page 5, line 187 does not make any sense….”…men have more risk factors for developing VTE than men.”
Response: According to the reviewer's suggestion errors in the text have been corrected in the revision.
- Query: Line 207-208 on page 5, the word “various” is used three times in the same sentence. This should be improved.
Response: According to the reviewer's suggestion errors in the text have been corrected in the revision.
- Query: Line 213 on page 5, the word “zero” should be “blood group O”.
Response: According to the reviewer's suggestion errors in the text have been corrected in the revision.
- Query: On page 7, lines 333 and 334, I did not understand the “o” placed in front of “genetic deficiency…..”, increase in plasma……”, and “persistent increase in……”. Please explain.
Response: According to the reviewer's suggestion errors in the text have been corrected in the revision.
- Query: Line 374, page 8, there is an extra “.” in the line after the word “used”.
Response: According to the reviewer's suggestion errors in the text have been removed in the revision.
Best regards
Tiziana Ciarambino
Round 2
Reviewer 2 Report
My only comment remains on line 198, group "0" should be group "O". Otherwise the authors have appropriately answered my concerns.
Author Response
Naples, 07 December 2022
Dear Prof. Doctor
Please, find enclosed the revised version of the manuscript entitled: Gender Medicine: A New Possible Frontiers of Venous Thromboembolism. We thank the Editor and the Reviewers for their comments and we hope that the following changes will now make the manuscript suitable for publication on the Stresses. Please see the following list of the changes made in manuscript.
Reviewer 2.
- Query: Line 198 on page 5, the word “0” should be “blood group O”.
Response: According to the reviewer's suggestion errors in the text have been corrected in the revision.
Best regards
Tiziana Ciarambino